# Comparative Evaluation of the In Vitro Cytotoxicity of a Series of Chitosans and Chitooligosaccharides Water-Soluble at Physiological pH

**DOI:** 10.3390/polym15183679

**Published:** 2023-09-06

**Authors:** Catia Dias, Loris Commin, Catherine Bonnefont-Rebeix, Samuel Buff, Pierre Bruyère, Stéphane Trombotto

**Affiliations:** 1UPSP 2021.A104 ICE, Interaction Cellule Environnement, VetAgro Sup, Université de Lyon, F-69280 Marcy l’Etoile, France; loris.commin@vetagro-sup.fr (L.C.); catherine.bonnefont@vetagro-sup.fr (C.B.-R.); samuel.buff@vetagro-sup.fr (S.B.); pierre.bruyere@vetagro-sup.fr (P.B.); 2Univ Lyon, CNRS, UMR 5223, Ingénierie des Matériaux Polymères, Université Claude Bernard Lyon 1, INSA Lyon, Université Jean Monnet, F-69622 Villeurbanne, France; stephane.trombotto@univ-lyon1.fr

**Keywords:** chitosan, COS, cytotoxicity, solubility

## Abstract

Chitosans (CS) have been of great interest due to their properties and numerous applications. However, CS have poor solubility in neutral and basic media, which limits their use in these conditions. In contrast, chitooligosaccharides (COS) have better solubility in water and lower viscosity in aqueous solutions whilst maintaining interesting biological properties. CS and COS, unlike other sugars, are not single polymers with a defined structure but are groups of molecules with modifiable structural parameters, allowing the adaptation and optimization of their properties. The great versatility of CS and COS makes these molecules very attractive for different applications, such as cryopreservation. Here, we investigated the effect of the degree of polymerization (DP), degree of N-acetylation (DA) and concentration of a series of synthesized CS and COS, water-soluble at physiological pH, on their cytotoxicity in an L929 fibroblast cell culture. Our results demonstrated that CS and COS showed no sign of toxicity regarding cell viability at low concentrations (≤10 mg/mL), independently of their DP and DA, whereas a compromising effect on cell viability was observed at a high concentration (100 mg/mL).

## 1. Introduction

Over the last two decades, numerous studies have revealed the interesting properties, namely the biocompatibility, biodegradability and biological activity, of chitosan (CS) [1,2]. CS is a natural linear polysaccharide derived from chitin, which is mainly found in arthropod exoskeletons and cephalopod endoskeletons [3]. CS is a co-polymer composed of N-acetyl-D-glucosamine (GlcNAc) and D-glucosamine (GlcN) units linked by β-(1→4) glycosidic bonds. Its physical, chemical and biological properties are partly defined by its degree of polymerization (DP), defined as the average number of GlcN and GlcNAc units in the CS macromolecules, and its degree of N-acetylation (DA), corresponding to the molar ratio of GlcNAc units [4,5,6]. Chitosan can be processed in various physical forms, such as gels, nanoparticles, films or composite materials [7,8,9,10]. It has therefore been studied in numerous applications, including food, cosmetics, agriculture and the biomedical field [11,12,13,14,15]. However, the use of CS is often hindered by its high viscosity in dilute aqueous acid solutions or its poor solubility at neutral and basic pH. Consequently, growing interest has been shown in chitooligosaccharides (COS), defined as oligomer forms of chitosan or chitin, with a DP less than 20, i.e., with an average molar mass lower than 4000 g/mol [16,17]. Compared to chitosan, COS show better water solubility and lower viscosity in aqueous solutions. Furthermore, COS are also known to have specific biological properties, such as antifungal, antibacterial or immuno-enhancing effects on animals [18]. COS have also been shown to elicit increasingly protective responses in various plants and possess antimicrobial activity against a wide spectrum of phytopathogens [19].

The great versatility of CS and COS could make these molecules very attractive for applications in the field of cryopreservation. The development of cryopreservation methods is constantly progressing to achieve higher survival rates and the preservation of biological functions. In fact, the long-term preservation of cells and tissues is essential in many scientific fields, from assisted reproduction and biomedical research to animal and plant biodiversity conservation [20,21,22,23]. The cryopreservation of embryos has significant economic and genetic benefits and, combined with embryo transfer, has contributed to the global distribution of reproductive material, replacing the live animal trade [24]. Nevertheless, cryopreservation methods require cryoprotectant agents (CPAs) (such as ethylene glycol, dimethyl sulfoxide, propanediol, etc.), most of which are relatively toxic to cells. This is particularly true for vitrification, which requires high concentrations of penetrating CPAs. Embryo vitrification is a cryopreservation technique that consists of a transition from a liquid state to a glassy state, without crystallization. This state is achieved by using very high concentrations of CPAs, which induce very high viscosity in the medium, as well as ultra-rapid cooling–warming rates, which are necessary to prevent the formation of ice crystals [20,25,26]. Penetrating CPAs in vitrification solutions, despite considerable improvements over the years, remain a major concern due to their toxicity [21,27,28]. Among the various molecules already studied, sugars such as sucrose, trehalose or glucose have been proposed to limit the use of penetrating CPAs. These molecules demonstrate low toxicity, increase the viscosity of the vitrification solution and promote cell dehydration and glass formation, without affecting the vitrification properties [27,29,30,31]. In contrast to other sugars, the DP and DA of CS and COS can be modified, allowing their properties and therefore their cryoprotective behavior to be adapted and optimized, making CS and COS good alternatives to replace all or part of the penetrating cryoprotectant in vitrification solutions.

Nevertheless, the use of CS and COS in vitrification solutions requires potentially much higher concentrations compared to their conventional use. In fact, we assume that the more molecules there are in the solution, the more hydroxyl groups will be available to form hydrogen bonds with water, making water less available to form ice crystal bonds, so less water will be available to form ice crystals. In addition, the vitrification process will be enhanced due to the higher concentrations of molecules and consequently higher viscosity. Consequently, an evaluation of the biocompatibility of CS and COS is necessary to study their potential cytotoxicity at high concentrations.

The aim of the present study was therefore to evaluate the effect of the structural parameters (DP, DA) and the concentrations of a series of CS and COS, water-soluble at physiological pH, on their cytotoxicity in L929 fibroblasts, a cell line recommended by international standard procedure ISO 10993-5 [32].

## 2. Materials and Methods

Commercial shrimp CS 244LG (batch 20140503; DA ~1%; Mw = 201.3 kg/mol; Mn = 118.7 kg/mol; Ð = 1.696) were provided by Mahtani Chitosan Ltd. (Veraval, India). Sodium nitrite (NaNO_2_, purity > 99%), hydrochloric acid (37% *w*/*w*), deuterium oxide (purity > 99.96% atom D), ammonium hydroxide (28% NH_3_ in water, purity > 99.9%), glacial acetic acid (purity > 99.7%) and acetic anhydride (purity > 99.5%) were provided by Sigma-Aldrich (Saint-Quentin Fallavier, France).

The CS and COS synthetized in this study are, respectively, referred to as CS_DP/DA_ or COS_DP/DA_ according to their DP and DA values.

### 2.1. Preparation of COS with Low DA (<1%) by Nitrous Acid Depolymerization of Chitosan

These COS were prepared according to the method described by Moussa et al. [16]. Thus, commercial chitosan 244LG (2 g, 12 mmol of GlcN unit) was solubilized in 100 mL of water by the addition of 1.2 mL of HCl (37% *w*/*w*). A freshly prepared 5 mL aqueous solution of NaNO_2_ (GlcN/NaNO_2_ molar ratio = 3.5 for rCOS_17/1_ (expected for reacetylation), COS_17/1_ and COS_22/0_, =9 for COS_22/1_, =20 for COS_35/1_ and =75 for CS_122/2_) was added and the mixture was stirred for 12 h at ambient temperature. Then, sodium hydroxide (1 M) was added to the solution to reach a neutral pH before reduction with sodium borohydride (NaBH_4_). NaBH_4_ (12 mmol) was added to the solution at a temperature below 10 °C; then, the solution was stirred for 12 h to ensure the total reduction of the free aldehyde group at the reducing end unit and the better long-term chemical stability of COS in the aqueous solution [33,34]. For rCOS_17/1_, COS_17/1_ and COS_22/0_, HCl (1 M) was added to the solution to reach a neutral pH, and then the solution was filtered (0.45 μm Millipore CME membrane) and concentrated by vacuum evaporation. After precipitation in acetone, several washings of the precipitate with methanol/acetone (1:1 *v*/*v*) were performed. The powder was solubilized in water, dialyzed with a cellulose membrane (MWCO 100–500 Da) and finally freeze-dried. After the filtration of the solution, COS_35/1_ and CS_122/2_ were directly precipitated by the addition of NH_3_ (28% *w*/*w*) (NaOH for COS_22/1_) to pH 8–9, washed several times with deionized water until a neutral pH was reached and then freeze-dried. All COS samples were obtained as a white powder with a mass yield from 70 to 80%.

### 2.2. Preparation of CS and COS with High DA by Acetic Anhydride Reacetylation

#### 2.2.1. Preparation of COS with High DA (From 35% to 57%)

These COS were prepared based on the studies of Abla et al. [35]. Thus, for the preparation of COS22/52, COS22/1 (1 g) was solubilized in 100 mL of water/ethanol (1:1 *v*/*v*) and glacial acetic acid (0.4 mL). At a temperature below 10 °C, fresh acetic anhydride (0.32 mL) was added dropwise in a stoichiometric amount versus GlcN units to obtain the expected DA. After 12 h of stirring at ambient temperature and the filtration (0.45 μm CME membrane) of the solution, NH_3_ (28% *w*/*w*) was added to reach pH 8–9; then, the solution was evaporated under a vacuum. The precipitate was thoroughly washed with ethanol, followed by several washings with acetone. After drying under a vacuum, the product was solubilized with deionized water, dialyzed with a cellulose membrane (MWCO 100–500 Da) and finally freeze-dried (63% mass yield). Using the same procedure, COS_17/51_ and COS_18/35_ were synthesized from rCOS_17/1_ and COS_36/57_ from COS_35/1_ with mass yields around 60–70%.

#### 2.2.2. Preparation of CS with High DA (~50%)

These CS were prepared according to the method described by Lamarque et al. [36]. Thus, for the preparation of CS_984/50,_ CS 244LG (1 g) was solubilized in 100 mL of water/propane-1,2-diol (1:1 *v*/*v*) and glacial acetic acid (0.4 mL). At a temperature below 10 °C, fresh acetic anhydride was added dropwise in a stoichiometric amount versus GlcN units to obtain the expected DA. After 12 h of stirring at ambient temperature and the filtration (0.45 μm CME membrane) of the solution, CS_984/50_ was precipitated by the addition of NH_3_ (28% *w*/*w*) to reach pH 8–9, washed several times with deionized water until a neutral pH was reached and then freeze-dried (55% mass yield). Using a similar procedure, CS_100/49_ was prepared from COS_122/2_. After 12 h of stirring at ambient temperature and the filtration of the solution, it was concentrated by vacuum evaporation. Then, the product was precipitated by acetone, and the precipitate was thoroughly washed with acetone and then dried under vacuum. Finally, the precipitate was solubilized in deionized water and dialyzed (MWCO 1 kg/mol) before freeze-drying (80% mass yield).

### 2.3. Characterization Methods of CS and COS

#### 2.3.1. Proton Nuclear Magnetic Resonance Spectroscopy (^1^H NMR)

The DA of CS and COS samples was determined by ^1^H NMR according to the method described by Hirai et al. [37]. ^1^H NMR spectra were recorded on an AV300 Bruker (300 MHz) spectrometer at ambient temperature (Bruker, Billerica, MA, USA). All samples were dissolved at 10 mg/mL in D_2_O with 5 μL HCl (12 N) and transferred to 5 mm NMR tubes. Trimethylsilyl-3-propionic-2,2,3,3-D4 acid sodium salt (TMPSA, Sigma-Aldrich, Saint-Quentin Fallavier, France) was used as an internal reference. The Bruker Topspin software was used for the analysis of spectra (Bruker, version 3.6).

#### 2.3.2. Size Exclusion Chromatography (SEC)

The average molar masses (Mw and Mn) and the dispersity Đ of CS and COS samples were determined by size exclusion chromatography (SEC). CS and COS samples were first dissolved in a 0.2 M acetic acid/0.15 M ammonium acetate buffer (pH 4.5) at a concentration from 0.5 to 5 mg/mL according to the DP, for a minimum of 18 h at ambient temperature. The solutions were then filtered using 0.45 μm pore size CME membranes (Millipore, Burlington, MA, USA). The macromolecule separation was performed on two serially connected columns (TSK G2500PW and TSK G6000PW, Tosoh Bioscience, Tokyo, Japan). A differential refractometer (Optilab T-rex, Wyatt Technology, Santa Barbara, CA, USA) coupled online with a MALLS detector (Dawn Heleos II, Wyatt Technology, Santa Barbara, CA, USA) was used for the detection. A degassed 0.2 M acetic acid/0.15 M ammonium acetate buffer (pH 4.5) was used as an eluent after filtration on a 0.10 μm pore size membrane (Millipore). The flow rate was maintained at 0.5 mL/min, and the amount of sample injected was 50 μL. The refractive index increment (dn/dc) was adjusted for each acetylation degree (DA) according to the results of Schatz et al. [38]. The ASTRA 6.1 software (Wyatt Technology) was used for the analysis of chromatograms.

#### 2.3.3. Thermogravimetric Analysis (TGA)

Water and ash content of CS and COS samples was determined by TGA using an SDT-Q600 analyzer (TA Instruments, New Castle, DE, USA). Thermogravimetric analyses were performed under a flow of air (60 mL/min), with 20–30 mg of each sample, and by using a temperature ramp of 2 °C/min from ambient temperature to 200 °C before an isotherm of 15 min at 200 °C, a temperature ramp of 20 °C/min up to 900 °C and finally an isotherm of 45 min at 900 °C. The TA Universal Analysis 2000 software was used for the analysis of thermograms (TA Instruments, version 4).

### 2.4. L929 Cell Culture

Mouse fibroblasts of the L929 cell line (Sigma-Aldrich) were used to evaluate the cytotoxicity of the different CS and COS molecules. Cells were cultured in culture flasks in the complete culture medium Dulbecco’s Modified Eagle’s Medium (DMEM, Eurobio Scientific) supplemented with 10% fetal calf serum (Eurobio Scientific, Les Ulis, France), 1% penicillin/streptomycin (Eurobio Scientific), 1% L-glutamine (Eurobio Scientific) and 0.1% amphotericin B (Eurobio Scientific). Cells were incubated at 37 °C in a humidified atmosphere containing 5% CO_2_. After 7 days of culture, cells were harvested at approximately 80% confluence by treatment with 0.5 g/L trypsin–0.2 g/L EDTA (Eurobio Scientific). Cell cultures were tested by PCR to determine the absence of mycoplasma.

### 2.5. In Vitro Standard Cytotoxicity Test

The standard cytotoxicity test was performed according to the international standard procedures ISO 10993-5 [32] and ISO 10993-12 [39], as follows.

L929 cells were seeded at a density of 1.0 × 10^4^ cells/well in a 96-well plate at 37 °C in a humidified atmosphere containing 5% CO_2_. After 24 h of culture, the complete culture medium was replaced with fresh medium containing different concentrations of COS (COS_17/1_, COS_17/51_, COS_18/35_, COS_22/0_, COS_22/52_, COS_36/57,_ CS_100/49_ and CS_984/50_). A range of concentrations, from low concentrations to the solubility threshold (~100 mg/mL) of each synthetized molecule, was tested: 0.1, 1, 10 and 100 mg/mL. The assay was performed in triplicate (biological replicate), with each assay performed in three wells for each sample type (technical replicate). Negative controls (no cytotoxic response, according to ISO 10993) consisted of the cell culture medium and positive controls (cytotoxic response) were dimethyl sulfoxide (DMSO, Sigma-Aldrich, Saint Quentin Fallavier, France). After 24 h of incubation, the cells were rinsed with fresh cell culture medium and 10 μL of Cell Counting Kit-8 (CCK8, Sigma Aldrich) was added to each well and incubated for 2 h (37 °C, 5% CO_2_) to determine cell viability. Absorbance was measured at 450 nm using a microplate reader (MultiSkan, Thermofisher, Waltham, MA, USA). Cell viability was expressed as the percentage of absorbance relative to the control group (cells not exposed to the CS and COS solutions). A decrease in cell viability of more than 30% was considered as a cytotoxic effect (no cytotoxic effect for cell viabilities > 70%).

### 2.6. Osmolality of Solutions

The osmolality of each solution tested was determined by measuring the dew point depression of the sample with a vapor pressure osmometer (Vapro, Elitech France^®^). Each solution was tested in triplicate.

### 2.7. Statistical Analysis

For the cytotoxicity evaluation, viability results were expressed as mean ± SEM and differences between groups were tested using the Kruskal–Wallis non-parametric test and Bonferroni–Holm’s post-test for group comparisons. A *p*-value less than 0.05 was considered significant. All graphs and statistical analyses were performed using the GraphPad Prism software (Prism, version 5).

## 3. Results

### 3.1. Structural Characterizations of Synthesized COS and CS and Determination of Their Osmolality

Different CS and COS samples covering a wide range of DP from 17 to 984 and DA from 1% to 57% were synthesized. For each sample, the combination of DP and DA values was chosen in order to obtain chitosan compounds soluble at physiological pH. Thus, COS with low DA (<1%) were produced by nitrous acid depolymerization from a commercial low-DA chitosan according to the procedure described by Moussa et al. [16]. COS and CS with high DA (from 35% to 57%) were prepared by acetic anhydride reacetylation from low-DA COS and CS, respectively, based on the studies of Abla et al. [35] and Lamarque et al. [36].

In order to fully characterize these samples, several analysis techniques including SEC, ^1^H NMR and TGA were performed. Chemical parameters, such as DP, Mn, Mw and Ð determined by SEC, DA determined by ^1^H NMR and contents of water and ash determined by TGA, are given in Table 1. All size exclusion chromatograms, ^1^H NMR spectra and TGA thermograms can be found in the Appendix A (see Appendix A).

The osmolality of DMEM was used as a control. At low concentrations (<10 mg/mL), the osmolality remained close to the control. The higher concentration of 100 mg/mL for COS_17/1_, COS_22/0_, COS_18/35_ and COS_17/51_ showed a remarkable effect on osmolality, significantly increasing the osmolality of the solution (Table 2).

### 3.2. Cytotoxicity Analysis

The results of the in vitro cytotoxicity studies were obtained after 24 h of incubation with different concentrations of CS and COS from 0.1 to 100 mg/mL, as shown in Figure 1 and Figure 2. The potential toxic effect of CS and COS was evaluated as a function of DP, DA and concentration.

#### 3.2.1. Influence of the Concentration

Our results showed that the same pattern was observed for all CS and COS samples over the entire concentration range (Figure 1).

As shown in Figure 2, at low concentrations (≤10 mg/mL), cell viability remained above the established 70% threshold (according to the international standard methods ISO 10993-5 and ISO 10993-12), which corresponded to the absence of a cytotoxic effect. There was no significant difference between DMEM and the various COS and CS at these low concentrations. Conversely, a cytotoxic effect of CS and COS was observed at concentrations of 100 mg/mL (and 50 mg/mL for the CS_100/49_).

#### 3.2.2. Influence of DP and DA

To evaluate the influence of DP, we compared CS and COS with the same DA and different DP (Figure 3). The molecules used for this comparison were COS_17/51_, COS_22/52_, COS_36/57_, CS_100/49_ and CS_984/50_. All molecules had the same DA (~50%) and a DP ranging from 17 to 984. Our data showed that cell viability was not affected by the DP but was concentration-dependent (Figure 1). Cell viability showed a decrease when exposed to a concentration higher than 10 mg/mL but was not affected at lower concentrations.

The same principle was followed to evaluate the DA (Figure 3). All molecules with the same DP and a different DA were compared. For COS with DP ~ 17 and DA ranging from 0 to 50% (COS_17/1_, COS_18/35_ and COS_17/51_), no effect of DA on cytotoxicity was observed, but rather a concentration effect, as observed for the effect of the DP.

## 4. Discussion

Despite extensive studies on the biological applications of CS and COS, there is no consensus on their biocompatibility. Given the diversity of their existing chemical structures (differences in molar mass, DP and DA) and the variety of related types of biological activity described, the comparison of the literature data on the cytotoxicity of CS and COS is a complex task. The cytotoxicity of CS has been repeatedly reported as a function of its structural properties such as Mw and consequently its DP, DA and concentration [2,40,41]. In this study, we designed a cytotoxicity study on a panel of CS and COS covering a wide range of DP and DA. In order to synthesize this panel, a choice was made between the various existing synthesis methods. Physical, enzymatic and chemical methods have been described for the preparation of low-molar-mass CS. In chemical methods, several acids can be used for the depolymerization of CS [42]. We chose nitrous acid (HNO_2_) as nitrous acid deamination is a well-known CS depolymerization method that offers several advantages. First, this reaction can be performed in an aqueous solution under mild temperature and acidity conditions. Moreover, this homogeneous reaction is specific to GlcN units and the number of glycosidic bonds broken is roughly stoichiometric to the amount of nitrous acid used, leading to the good control of DP [34,43]. For the good control of DA, we used acetic anhydride as the acetylating agent for the reacetylation of CS and COS, since acetic anhydride has shown the best acetylation efficiency in aqueous media [35]. To clarify the influence of the DP, DA and concentration on the toxicity of these molecules, we evaluated the effect of CS and COS with different DP and DA over a wide range of concentrations: from low concentrations (0.1 mg/mL) to the solubility threshold (from 10 to 100 mg/mL) of each CS and COS molecule on cell viability.

### 4.1. Influence of the Concentration

Our experimental data suggested that CS and COS were not deleterious to cell survival and development at concentrations below 10 mg/mL, as observed in other studies [44,45]. These results showed that the use of these low concentrations of COS and CS could be associated with high cell survival rates (>70% compared with control cells), which was consistent with also the literature. A previous study showed that CS (Mw 50–190 kg/mol, DA ~ 15%) is not toxic to normal L929 fibroblast cells, at concentrations of 500 µg/mL [45]. Another study described a low impact of COS (DP/DA of 10/53, 24/24, 24/47 and 45/47) on the cell viability of fibroblasts, at 10 mg/mL, after 48 h of incubation [44].

Conversely, cell viability was strongly impacted by a concentration of 100 mg/mL. One explanation could be linked to the osmolality of the solution. At low concentrations (<10 mg/mL), the osmolality of the CS and COS solutions remained close to the control, which was consistent with the cell viability. At higher concentrations (100 mg/mL), the osmolality of the media was affected for different COS and CS and consistent with the observed undesirable effects on cell viability. When comparing CS_100/49_ at 50 mg/mg and CS_984/50_ at 10 mg/mL, we observed that their osmolality was similar (355.5 and 358.4 mmol/kg, respectively), although the viability rates were different (69% vs. 92% for CS_100/49_ and CS_984/50_, respectively). Osmolality therefore seemed to be able to explain only part of the cell survival, and the intrinsic toxicity of the different chitosan molecules could be suspected for high concentrations. Moreover, Schimpf et al. reported that COS (Mw 1.4 kg/mol and DA ~ 22%) at 50 mg/mL was not cytotoxic to human spermatozoa [46]. However, the incubation time (30 min) in their study was shorter compared to ours (24 h), which could explain the difference. Unsurprisingly, the exposure time was also a parameter that could affect the toxicity of CS and COS. Finally, we noticed that amongst the synthesized COS and CS used in this study, only COS_17/1_, COS_22/0_, COS_18/35_ and COS_17/51_ significantly increased the osmolality of the solution at 100 mg/mL, which could be explained by the low DP of these molecules (DP ~ 17–18).

### 4.2. Influence of DP and DA

Our results indicated that the cytotoxic effect of CS and COS was affected by the concentration, not the DP. Similar results were obtained by Mao et al. [47], who investigated the relationship between Mw (Mw ranging from 5 to 400 kg/mol and DA ~ 15%) of CS and their cytotoxicity for L929 cells. They observed that the cytotoxicity of a CS was independent of its Mw but was dependent on its concentration, with a toxic effect at a concentration starting from 1 mg/mL. Several studies have been performed to clarify the relationship between Mw and cytotoxicity, but the results are controversial. Fernandes et al. [48] observed that COS (Mw 1.8–4.1 kg/mol and DA 30–35%) could exert strong cytotoxic effects, whereas CS (Mw 125.6 kg/mol, DA ~ 35%) showed significantly less toxicity. Chae et al. [49] reported that cell viability was significantly affected by the concentration and the Mw of CS and COS. At a low concentration (<1 mg/mL), there was no cytotoxic effect on Caco-2 cells. With increasing concentrations, the cytotoxic effect of CS was seriously affected by its Mw. A significant increase in the cytotoxicity of CS with a Mw of 230 kg/mol (DA ~ 15%) and 22 kg/mol (DA ~ 11%) was observed compared to a Mw of 3.8 kg/mol (DA ~ 12%) for concentrations higher than 5 mg/mL. In our study, in order to evaluate the influence of DP (and consequently the Mw), the compounds had to be compared at concentrations up to 10 mg/mL, since two of them were not soluble at a concentration of 100 mg/mL (CS_100/49_ and CS_984/50_ are soluble up to 50 mg/mL and 10 mg/mL, respectively). Compared to Chae et al. [49], our data did not show the same negative effects on cell viability at a concentration of 10 mg/mL, although this study used a short incubation time (2 h) compared to our study (24 h). Regarding the influence of DA, our results suggest that DA was not associated with any particular cytotoxicity. All COS molecules showed significant cytotoxicity only at high concentrations (100 mg/mL), whilst their cytotoxicity was not reduced by the modification of their DA. Huang et al. described that DA had a more important effect on the cytotoxic profile of CS than Mw [50]. They indicated that cell viability was seriously affected by the concentration and DA of CS and also showed that, at concentrations higher than 0.74 mg/mL, cell viability was significantly affected regardless of the Mw of CS. However, increasing the DA from 12% to 39% attenuated its cytotoxic effect for CS of Mw 213 kg/mol. Schipper et al. claimed the opposite conclusions, stating that the toxicity of CS seemed to be related to the DA, since CS of Mw 12–190 kg/mol showed more toxic effects for DA < 35% [51]. In contrast, our results did not show the same influence, which could potentially be explained by the difference in the tested molecules’ Mw: 213 kg/mol in Huang et al. [50], 12–190 kg/mol in Schipper et al. [51] and 3.2–5.1 kg/mol in our study.

## 5. Conclusions

In this study, we aimed to clarify the biocompatibility of CS and COS molecules, especially at high concentrations. For this purpose, we performed for the first time a 24-h international standard cytotoxicity test, testing different molecules of COS and CS, varying by their DP and DA, in a wide range of concentrations, covering low to very high concentrations. We demonstrated that neosynthesized CS and COS showed no sign of toxicity regarding cell viability at low concentrations (≤10 mg/mL), independent of their DP and DA. However, CS and COS showed a compromising effect on cell viability at 100 mg/mL, which provides an indication of potential toxicity at this concentration. Clearly, these results need to be considered depending on the application. It may be possible to overcome the toxic effect of high concentrations of COS and CS by reducing the incubation time, which could be very interesting in the field of cryopreservation, especially in vitrification applications, since the incubation time of the cells with the solutions is very limited (less than 10 min). Thus, the exposure time and cell model seem to play an important role and should be carefully considered in future work. In conclusion, these data could complement the currently available data to elucidate the toxicity of COS and CS molecules and highlight potential candidate molecules for further cryopreservation applications. Further studies are recommended to better characterize the cytotoxicity of CS and COS.

## Figures and Tables

**Figure 1 polymers-15-03679-f001:**
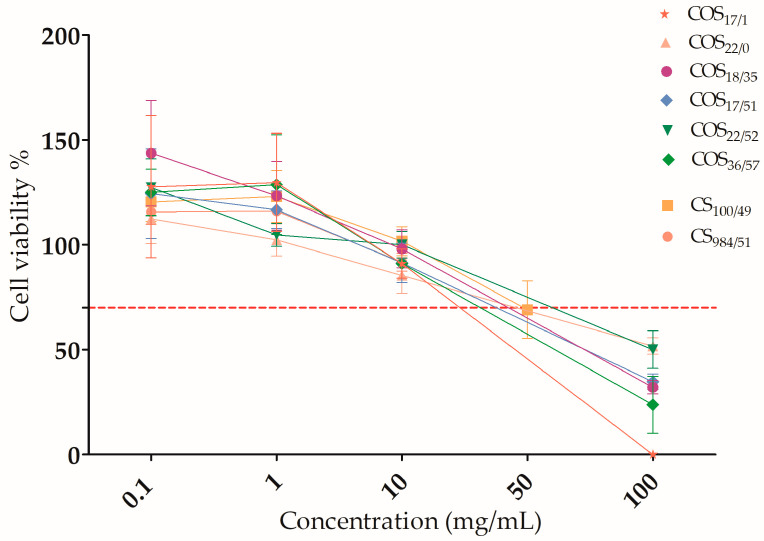
Cell viability assay. Cell viability of L929 was tested at different concentrations of CS and COS (0.1–100 mg/mL) after 24 h incubation. Red dashed line corresponds to the threshold for cell viability (ISO 10993-5). Cell viability is expressed as mean ± SEM.

**Figure 2 polymers-15-03679-f002:**
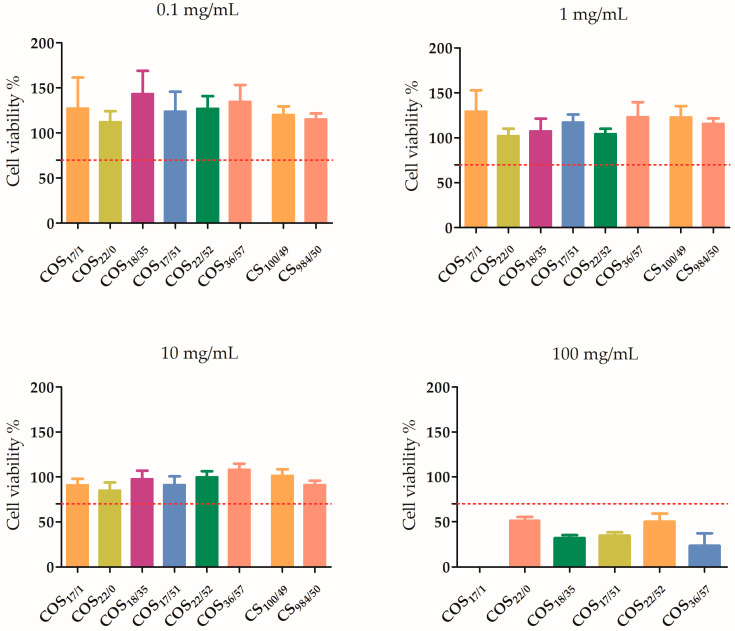
Cell viability assay. Cell viability of L929 was tested at different concentrations of CS and COS (0.1–100 mg/mL) after an incubation period of 24 h. Red dashed line corresponds to the threshold for cell viability (ISO 10993-5). Cell viability is expressed as mean ± SEM. COS17/1 show cell viability of 0%.

**Figure 3 polymers-15-03679-f003:**
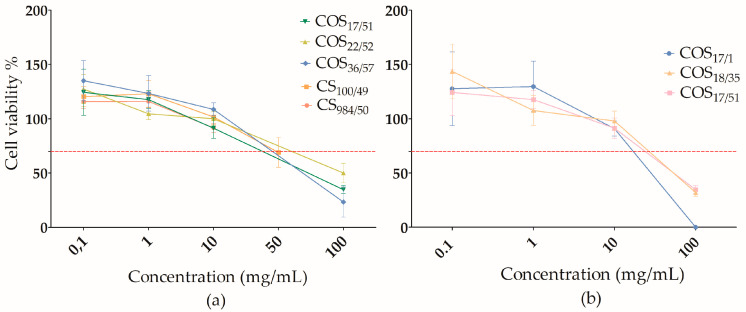
Cell viability assay. Cell viability of L929 was tested with CS and COS with different DP and DA: (**a**) solutions of CS and COS with different DP and DA around 50% and (**b**) solutions of COS with different DA and DP around 17, after an incubation period of 24 h. The red dashed line shows the cell viability threshold (ISO 10993-5). Cell viability is expressed as mean ± SEM.

**Table 1 polymers-15-03679-t001:** Characterization of synthesized chitosans and chitooligosaccharides.

Sample	DP ^a^	DA (%) ^b^	Mn (kg/mol) ^a^	Mw (kg/mol) ^a^	Ð ^a^	Water Content (% *w*/*w*) ^c^	Ash Content (% *w*/*w*) ^c^	Solubility Threshold (mg/mL)
COS								
COS_17/1_	17	1	2.68	3.19	1.19	5.1	1.3	100
COS_22/0_	22	0	3.53	5.08	1.44	6.4	0	100
COS_18/35_	18	35	3.09	4.97	1.61	9.8	0.5	100
COS_17/51_	17	51	3.17	5.05	1.60	6.5	0.4	100
COS_22/52_	22	52	3.94	5.17	1.31	16.5	0	100
COS_36/57_	36	57	6.60	12.1	1.83	11.6	0	100
CS								
CS_100/49_	100	49	18.3	25.6	1.40	10.2	0.8	50
CS_984/50_	984	50	179	318	1.77	12.9	0	10

^a^: Number average degree of polymerization (DP), number average molar mass (Mn), mass average molar mass (Mw) and dispersity (Ð) were determined by SEC. DP was calculated as Mn/M_0_ with M_0_ = (DA × M_GlcNAc_ + (100 − DA) × M_GlcN_)/100, M_GlcNAc_ = 203 g/mol and M_GlcN_ = 161 g/mol; ^b^: The average degree of N-acetylation (DA) was determined by ^1^H NMR. ^c^: Contents of water and ash were determined by TGA. Chitosans (CS) and chitooligosaccharides (COS) samples with different DP and DA are abbreviated as CS_DP/DA_ and COS_DP/DA_, respectively.

**Table 2 polymers-15-03679-t002:** Osmolality of CS and COS solutions (mmol/kg).

Solution	Concentration (mg/mL)					
	0	0.1	1	10	50	100
DMEM	341.2 ± 25.3					
DMEM + COS_17/1_		323.1 ± 12.0	323.1 ± 12.0	349.9 ± 38.0		431.6 ± 48.0 *
DMEM + COS_22/0_		320.3 ± 14.6	325.2 ± 11.6	349.4 ± 10.3		479.8 ± 62.2 ***
DMEM + COS_18/35_		320.4 ± 10.7	324.7 ± 12.5	359.1 ± 36.7		465.4 ± 18.6 ***
DMEM + COS_17/51_		325.3 ± 11.8	325.7 ± 10.8	366.0 ± 32.4		455.2 ± 16.4 **
DMEM + COS_22/52_		344.6 ± 30.3	332.1 ± 21.0	353.1 ± 14.4		387.0 ± 23.3
DMEM + COS_36/57_		319.1 ± 8.3	324.6 ± 8.2	329.0 ± 33.6		403.3 ± 47.5
DMEM + CS_100/49_		320.9 ± 25.1	330.9 ± 25.1	341.3 ± 14.2	355.5 ± 70.0 ^a^	
DMEM + CS_984/50_		321.4 ± 18.1	324.8 ± 11.1	358.4 ± 40.0		

^a^: CS_100/49_ was tested at 50 mg/mL as it was insoluble at 100 mg/mL. Data are mean ± S.D. (statistically significant at * *p* < 0.05, ** *p* < 0.01, *** *p* < 0.001). All groups are compared with the control DMEM.

## Data Availability

Not applicable.

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
