# Peer review of "Comparative Evaluation of the In Vitro Cytotoxicity of a Series of Chitosans and Chitooligosaccharides Water-Soluble at Physiological pH"

_polymers, 2023, doi:10.3390/polym15183679_

Round 1
Reviewer 1 Report
The article is well written. The samples of chitooligosaccharides are thoroughly characterized. The results and conclusions are clearly written and correspond to the objectives of the work.
In the authors of this paper, toxicity begins to appear around 5 mg/mL, which agrees with https://doi.org/10.3892/or.2014.3463, in this paper, where COSs at a concentration of 4 mg/mL showed toxicity and live cells remained less than 75% .
We would like to point out that in most articles, researchers most often study the cytotoxicity of chitosan at concentrations of 0.5 or 1 mg/mL (https://doi.org/10.3390/foods12142740, https://doi.org/10.1016/j.carres.2022.108678). In this work the concentration reaches up to 10 mg/mL, such high concentrations are probably related to good solubility of samples?
Have you come across any other works using a similar range of concentrations?
Why do the authors use L929 cell line, is there any explanation?
In my opinion, Table 1 with complete characteristics should be placed in the body of the article, as it contains important characteristics of the samples, there is no need to move it in the supplementary files.
Although the methodology is written in enough detail, sections 2.2.1 and 2.2.2 lack references, or is this methodology so unique that there is no need for references?
Probably the authors did not notice, but paragraph 2.2.1 should be called Preparation, not Reacetyltion.
Minor editing of English language required
Author Response
Dear reviewer,
Please see the attachement.

Reviewer 2 Report
The paper "Comparative evaluation of the in vitro cytotoxicity of a series of chitosans and chitooligosaccharides water-soluble at physiological pH" describes the influences of the polymerization degree, degree of N-acetylation and the concentration of various types of chitosans and chitooligosaccharides water-soluble on the cytotoxicity on L929 fibroblasts cell culture. I have some comments and suggestions for consideration to improve the manuscript quality:
1. I do not see the novelty in this study. The authors should better specify what is new in their work;
2. Table 1 is not necessary because is a small part of Table S1. In my opinion Table 1 should be removed and replaced with Table S1;
3. Figures S1-S24 are not specified in the text and no comments are made regarding the characterization of chitosan and chitooligosaccharides.
4. As the authors also said in the Conclusions Section, more studies are needed to elucidate the cytotoxicity of these products.
Author Response

(The authors gave the same response as above.)

Reviewer 3 Report
The manuscript is well written. The rationale of this study is neatly explained. Methods were described following standard references. Results were presented with enough data.
Author Response
Dear reviewer,
We would like to thank you very much for your comments.
Round 2
Reviewer 2 Report
The paper can be accepted for publication in present form.